# When Genes Wear Marks: Epigenomic Modulation in the Development and Progression of Obesity

**DOI:** 10.3390/ijms26168067

**Published:** 2025-08-20

**Authors:** Alexandra F. Nikolaeva, Marina V. Nemtsova, Anna V. Pustovalova, Vladimir O. Sigin

**Affiliations:** 1Research Centre for Medical Genetics, 115522 Moscow, Russia; a_pustovalova02@inbox.ru (A.V.P.); sigin.vladimir@gmail.com (V.O.S.); 2Institute of Molecular Medicine, I.M. Sechenov First Moscow State Medical University (Sechenov University), 119048 Moscow, Russia; nemtsova_m_v@mail.ru

**Keywords:** obesity, epigenetics, DNA methylation, microRNAs, chromatin modifications

## Abstract

Obesity represents a global medical and social challenge characterized by pathological accumulation of adipose tissue as a result of complex interactions among genetic, environmental, and epigenetic factors. Recent studies have highlighted the pivotal role of epigenetic mechanisms in the pathogenesis of this condition, including abnormal DNA methylation of metabolic genes, dysregulation of microRNAs, and chromatin remodeling. These modifications are reversible and can be modulated by dietary, behavioral, and pharmacological interventions. This review provides a comprehensive analysis of tissue-specific epigenetic alterations identified not only in adipocytes and hepatocytes but also in peripheral blood cells, offering promising opportunities for the development of novel diagnostic biomarkers and targeted epigenetic therapeutic strategies.

## 1. Introduction

Obesity is a multifactorial chronic disease that, in recent years, has exhibited a global prevalence affecting approximately 2.5 billion adults, corresponding to nearly 43% of the population aged 18 years and older [1]. Obesity is closely associated with the development of metabolic disorders, including cardiovascular diseases, type 2 diabetes, atherosclerosis, and oncological conditions, which impose a substantial burden both on affected individuals and healthcare systems [2]. The development of obesity in childhood is determined by both genetic and epigenetic factors. In a substantial proportion of children without identified genetic predispositions to early-onset obesity, the etiology of the disease may be associated either with as-yet unidentified genetic variants or with epigenetic modifications [3]. The contribution of hereditary factors to the pathogenesis of childhood obesity and the key genes associated with this condition continue to be actively investigated. In light of the increasing relevance of the issue, the role of epigenetic mechanisms in the development of early morbid obesity has become a subject of intensive scientific discussion [4].

Contemporary research on the epigenetics of obesity is based on three fundamental mechanisms: DNA methylation, which regulates the expression of key metabolic genes through modification of cytosines primarily located in CpG islands of gene promoters; microRNAs, which mediate post-transcriptional regulation of genes involved in adipogenesis and insulin signaling; and chromatin modifications, including histone marks and remodeling complexes that determine the accessibility of genomic loci (Figure 1) [5,6,7].

These interconnected systems establish stable patterns of gene expression that contribute to the development of metabolic disorders and represent promising targets for the diagnosis and treatment of obesity. These mechanisms can exert combined effects on specific cell types and tissues, both reflecting and responding to environmental influences. Environmental factors that promote obesity include disrupted eating patterns, abnormal dietary behavior, and physical inactivity. The interplay between environmental exposures and the host’s genetic background leads to epigenetic dysregulation in white adipose tissue (WAT), contributing to adipocyte hypertrophy and dysfunction, as well as impairments in other cell types.

A substantial body of evidence has been accumulated in the scientific literature regarding DNA methylation profiles in immune cells of peripheral blood and other adult tissues [8,9]. This review will also focus on the analysis of DNA methylation pattern alterations associated with severe early-onset morbid obesity, complementing the existing body of data. The relevance of this focus lies in the diagnostic challenges encountered in cases of early-onset obesity. While etiological therapies have been developed for monogenic forms of obesity, their prevalence within the overall structure of the disease remains relatively low, accounting for only 5.8% according to recent data [10]. Studies have shown that DNA methylation of certain genes correlates with changes in metabolic rate and the distribution pattern of adipose tissue. Moreover, associations have been identified between methylation patterns and the expression of genes involved in appetite regulation and lipid metabolism [11]. On the other hand, DNA methylation is characterized by pronounced tissue specificity, which necessitates the analysis of epigenetic modifications in tissues most relevant to the pathological process under investigation [12,13]. From the standpoint of studying epigenetic regulation in the context of excessive body weight, adipocytes of adipose tissue represent a key substrate for detailed analysis of DNA methylation changes [14].

MicroRNAs, as a critically important class of small non-coding RNAs that mediate post-transcriptional regulation of gene expression, exhibit substantial diagnostic potential as biomarkers of obesity and associated metabolic disorders. This is supported by significant differences in microRNA expression profiles between patients with obesity and type 2 diabetes and healthy donors, as demonstrated in large-scale scientific studies [15,16,17]. These molecules function as endocrine and paracrine mediators, facilitating intercellular and intertissue communication, and can be secreted by various tissues, including adipose tissue, thereby modulating metabolic processes in distant organs [18]. According to existing data, circulating microRNAs are characterized by a high degree of individual reproducibility and stability throughout the circadian cycle, as well as resistance to interference from other plasma components. Owing to their high sensitivity to standard detection methods, they represent a promising biomarker tool for investigating preclinical stages of metabolic diseases [19]. Numerous studies have confirmed the pivotal role of microRNAs in the pathogenesis of obesity through the regulation of fundamental biological processes, including the control of adipocyte differentiation via modulation of gene expression, the maintenance of energy homeostasis through the AMPK signaling pathway, the regulation of insulin sensitivity, and the induction of chronic low-grade inflammation by modulating the production of pro-inflammatory cytokines in adipose tissue macrophages [20,21,22,23].

The 3D organization of chromatin plays a critical role in the regulation of gene expression in obesity, as reflected by alterations in the spatial arrangement of chromosomal territories, the dynamics of topologically associated domain formation, and the reconfiguration of promoter–enhancer interactions. Studies have shown that obesity is associated with disruption of normal chromatin architecture in adipose tissue, leading to the dysregulation of major metabolic genes [24,25]. In adipocytes, obesity induces persistent epigenetic alterations that are maintained even after body mass index (BMI) reduction—a phenomenon referred to as memory of obesity. This is manifested by the sustained pro-inflammatory genes’ transcriptional profile and the disruption of normal promoter–enhancer interactions [26]. Notably, different adipose depots exhibit distinct patterns of chromatin organization and regulatory element accessibility, which may account for their differential susceptibility to metabolic disorders [27,28]. These alterations affect not only adipocytes but also immune cells within adipose tissue, contributing to the development of chronic inflammation through persistent changes in macrophage chromatin architecture [29]. Current evidence highlights that obesity is associated with profound remodeling of the three-dimensional genome organization, opening new avenues for the development of therapeutic strategies aimed at correcting these epigenetic abnormalities.

Thus, obesity develops under the influence of complex interactions among epigenetic mechanisms, including DNA methylation that regulates the expression of genes involved in metabolic pathways; microRNAs that control post-transcriptional processes; and chromatin remodeling that determines the spatial organization of the genome. These mechanisms, characterized by tissue specificity and persistence, establish pathological metabolic programs, underscoring their pivotal role in the pathogenesis of obesity and opening prospects for the development of novel diagnostic and therapeutic strategies.

## 2. DNA Methylation

DNA methylation, one of the most extensively studied epigenetic modifications, plays a key role in the regulation of gene expression, maintenance of genome stability, and control of development. In human cells, DNA methylation involves the covalent addition of a methyl group (–CH_3_) to the fifth carbon atom of the cytosine pyrimidine ring, predominantly occurring in the context of CpG dinucleotides [30]. The establishment and maintenance of DNA methylation patterns are mediated by a complex interplay among DNA methyltransferases (DNMTs), methyl-CpG-binding domain proteins (MBDs), histone modifications, and DNA demethylases involved in the removal of methyl groups. In gene regulatory regions, such as promoters and enhancers, the presence of DNA methylation is associated with gene expression repression. This transcriptional repression mechanism is executed through the formation of a condensed chromatin structure, which restricts access of the transcriptional machinery and suppresses gene expression [31]. Disruption of DNA methylation mechanisms is strongly associated with a broad spectrum of diseases, including obesity [32,33]. It should be emphasized that the establishment and maintenance of DNA methylation patterns is a dynamic and intricately regulated process, closely interconnected with other modes of epigenetic regulation. Dysregulation within the finely tuned mechanisms of DNA methylation and demethylation may constitute a major factor in the development of numerous pathologies.

### 2.1. Abnormal DNA Methylation in Immune System and Peripheral Blood Cells in Obesity

Peripheral blood represents a heterogeneous cell population comprising diverse immune and myeloid cells, each possessing a unique epigenetic profile. When total genomic DNA is extracted from peripheral blood cells followed by methylation profiling, the resulting epigenetic patterns predominantly reflect inflammatory processes in the body, due to the predominance of the leukocyte fraction in the cellular composition of peripheral blood. This phenomenon is explained by the fact that leukocytes, constituting the main cellular population in blood, exhibit specific methylation patterns associated with immune activation and inflammatory responses [34]. Existing scientific evidence indicates that obesity is accompanied by a low-grade chronic inflammatory process. In turn, obesity-induced chronic inflammation may exert a complex influence on neuroimmune interactions, affecting the blood–brain barrier, lymphatic system, and meningeal tissues. These alterations have the potential to disrupt the regulation of the body’s energy homeostasis [35].

Recent epigenetic studies have revealed significant differences in global DNA methylation status between patients with early-onset obesity and individuals with normal body weight. Analysis of peripheral blood samples demonstrated a statistically significant reduction in global DNA methylation levels in individuals with obesity compared to the control group. Notably, the study identified sex-specific differences in the degree of DNA hypomethylation: in male pediatric patients with obesity, global methylation levels were significantly lower than those observed in female patients with the same diagnosis [36].

A Korean study published in 2020 [37] identified differential DNA methylation patterns associated with obesity susceptibility in children. The most significant findings included hypermethylation of CpG cg11024682, located within the body of the *SREBF1* gene, and hypomethylation of cg13424229, situated in a GATA binding factor 1 cluster within the promoter region of *CPA3*, in patients with early-onset obesity. The *SREBF1* gene encodes sterol regulatory element-binding protein 1 (SREBP-1), a transcription factor that plays a role in the regulation of sterol biosynthesis, as well as glucose and lipid metabolism. *CPA3*, which encodes carboxypeptidase A3, is a recognized marker of inflammatory processes. This enzyme is involved in proteolytic processing and the degradation of peptide substrates. The authors demonstrated that hypomethylation of CpG cg13424229 correlated with increased *CPA3* expression. Promoter hypomethylation is typically associated with increased transcriptional activity of the corresponding gene, which is consistent with the observed *CPA3* expression levels in children with obesity. Modulation of this gene’s expression may potentially influence metabolic cascades involved in the pathogenesis of obesity.

In a study conducted by Lima, R. S. and colleagues [38], the primary focus was on the methylation analysis of specific cytokine genes, given the central role of inflammatory processes in the pathogenesis of obesity. The researchers identified hypermethylation of the *CXCL8* promoter region in peripheral blood samples from children with obesity compared to the control group. Observed differences in *CXCL8* methylation patterns between the studied groups suggest a potential influence of epigenetic modifications on gene expression, considering that promoter methylation is generally associated with transcriptional repression. *CXCL8*, also known as IL-8, acts as a mediator of immune cell migration to sites of inflammation and is regarded as a critical factor in the development of early-onset obesity. It is important to highlight the methodological rigor of this study: the authors applied exclusion criteria for individuals who had taken anti-inflammatory medications, thereby minimizing the potential impact of exogenous factors on the methylation analysis results. This approach enhances the reliability of the data obtained and their interpretation in the context of the relationship between epigenetic alterations and obesity.

In the context of early severe obesity research, *FTO* (alpha-ketoglutarate-dependent dioxygenase FTO) has attracted significant attention from the scientific community. Despite the established association of polymorphisms in this gene with obesity susceptibility, its precise physiological function remains a subject of debate. A 2021 study conducted on peripheral blood mononuclear cells from 16 patients with early-onset obesity provided new insights into the role of *FTO* in regulating metabolic processes. The results demonstrated increased methylation levels in a portion of the *FTO* promoter region in individuals with obesity compared to the control group [39]. Paradoxically, the average *FTO* expression level was also elevated in the obesity group. The authors hypothesized that increased methylation of the promoter region may impede the binding of the FTO protein to its own promoter. This could potentially disrupt the negative feedback mechanism, leading to the observed increase in *FTO* mRNA expression. This study contributes to the understanding of epigenetic regulatory mechanisms of *FTO* and its role in the pathogenesis of childhood obesity, as well as associated disturbances in glucose and lipid metabolism. However, the authors acknowledge a significant limitation of the study—the relatively small sample size—which may have affected the statistical power and reliability of the results. Further research on larger cohorts is necessary to validate these preliminary findings and to elucidate the mechanisms of epigenetic regulation of *FTO* in the context of early obesity.

A 2024 study [36] identified two significant differentially methylated CpG dinucleotides: the hypomethylated cg05831083 in the *TFAM* gene (mitochondrial transcription factor A), associated with obesity and vitamin D deficiency, and the hypermethylated cg14926485 in the *PIEZO1* gene (a mechanosensitive ion channel), which exhibited an opposite pattern. The findings suggest a potential role of epigenetic regulation of these genes in the pathogenesis of metabolic disorders, where *TFAM* hypomethylation may reflect impaired mitochondrial biogenesis, and *PIEZO1* hypermethylation may indicate disrupted mechanotransduction in adipocytes in the context of obesity.

It should be emphasized that cellular heterogeneity of whole blood samples can significantly influence the variability of DNA methylation profiles [40]. In this context, adjusting for cellular composition using computational deconvolution methods—such as the algorithm proposed by Houseman et al. in 2012 or similar approaches—represents a critical step in bioinformatics analysis [41]. Such an approach helps minimize potential systematic biases related to differences in the proportions of cellular subpopulations and enhances the reliability of the resulting epigenetic data.

### 2.2. Tissue-Specific Features of Abnormal DNA Methylation in Severe Obesity

In the context of DNA methylation studies in obesity, researchers are increasingly expanding the range of analyzed tissues beyond peripheral blood, focusing on those directly involved in the pathogenesis of the disease and its associated complications (Table 1). Key targets of investigation include subcutaneous (SAT) and visceral (VAT) adipose tissue, along with their cellular components—primarily mature adipocytes—given their direct involvement in the development and progression of obesity-related pathophysiology [42]. As demonstrated by recent studies, isolated analysis of epigenetic profiles in mature adipocytes provides significantly higher resolution compared to analyses of whole tissue samples [43,44]. This is attributed to the elimination of cellular heterogeneity effects caused by the presence of the stromal–vascular fraction, which contains preadipocytes, immune cells, and fibroblasts. Tissues selected for analysis are chosen based on their pathophysiological relevance to obesity and metabolic disorders, allowing for the identification of molecular changes directly associated with the development of these conditions [45]. In current studies of DNA methylation in extreme obesity, the analysis of adipocytes isolated from VAT and SAT plays a central role. Traditionally, visceral adipocytes have been considered to exert a more pronounced negative metabolic effect and to exhibit a distinct methylation profile under metabolic dysfunction [46,47]. However, the study by McAllan, L. et al. suggests that subcutaneous adipocytes exhibit a significantly greater number of CpGs associated with obesity [14]. In a cohort of patients with morbid obesity, global DNA hypomethylation was observed. Of particular interest is the hypomethylation of subcutaneous adipocytes with pronounced a signaling (endocrine) function in enhancer regions, flanking areas near active transcription start sites, and regions repressed by polycomb group (PcG) proteins. At the same time, these adipocytes exhibited hypermethylation in actively transcribed gene regions. In contrast, visceral signaling adipocytes showed reduced representation in enhancer regions. Functional screening of adipocytes revealed that during their differentiation, *PRRC2A* (Proline Rich Coiled-Coil 2A) is suppressed through methylation and reduced expression, leading to substantial lipid accumulation. Moreover, the role of *PRRC2A* methylation in adipogenesis in obesity is supported by a consistent decrease in the expression of *PARG*, a key regulator of adipogenic transcription factors. This, in turn, correlates with impaired lipid metabolism and the progression of insulin resistance.

Saliva is increasingly recognized as a valuable source of epigenetic information, complementing traditional blood and target tissue samples. This is particularly important for large-scale population studies and the development of non-invasive screening strategies aimed at identifying epigenetic biomarkers of preclinical disease stages [48,49,50]. Saliva represents a promising biological material for epigenetic studies, offering several significant advantages such as non-invasive sample collection, cost-effectiveness, and feasibility of obtaining specimens under various conditions. These qualities make saliva an especially valuable substrate for investigating epigenetic markers. Recent studies have demonstrated a substantial correlation between saliva and peripheral blood methylomes, supporting the potential validity of using saliva as a surrogate material for assessing systemic epigenetic alterations [51,52]. A recent Latin American study demonstrated the potential prognostic value of DNA methylation in salivary tissue for predicting the development of early-onset obesity [53]. Within a longitudinal study, saliva samples from children with overweight status were analyzed at baseline and again after 36 months in participants who developed obesity. The analysis revealed a statistically significant association between hypermethylation of CpG cg01307483 in *NRF1* and an increased likelihood of obesity development. These findings indirectly align with previously published results from a 2003 study conducted on peripheral blood and skeletal muscle samples from individuals with diabetes [54], which demonstrated reduced *NRF1* expression. The *NRF1* gene plays a crucial role in regulating innate immune response, adipocyte inflammation, cytokine expression, and thermogenic adaptation of brown adipose tissue [55,56,57]. The identified association highlights the potential role of *NRF1* epigenetic modifications in the pathogenesis of early-onset obesity. The authors of the study note that increasing the sample size may allow for the identification of additional CpGs associated with obesity risk and for achieving statistical significance for other potential markers. Furthermore, the researchers emphasize the need to conduct a similar analysis using peripheral blood samples to validate the findings and to assess the correlation between salivary DNA methylation and systemic biomarkers.

A recent study by Cantarero-Cuenca, A. et al. [58] demonstrated that DNA methylation profiles in VAT and peripheral blood show only partial correlation—accounting for 4.3% of the total analyzed CpGs. Nevertheless, a number of concordant epigenetic markers were identified, primarily associated with genes involved in metabolic processes and immune system function, including *HLA-DQ2*, which has been potentially linked to the development of metabolic syndrome. The authors suggest that the observed epigenetic alterations may reflect the influence of adipose tissue on immune cell activation through an imbalance of adipokines—namely, increased levels of pro-inflammatory leptin and decreased levels of anti-inflammatory adiponectin—as well as through endoplasmic reticulum stress and hypoxia in hypertrophied adipocytes. These factors contribute to the activation of inflammatory signaling pathways and subsequent immune dysregulation.

Thus, the analysis in the current research highlights the complex relationship between altered DNA methylation across various tissues and the pathogenesis of obesity. Specific epigenetic markers identified in immune cells and peripheral blood reflect systemic inflammation and metabolic disturbances characteristic of early-onset obesity. At the same time, tissue-specific studies—particularly analyses of methylation in adipocytes from VAT and SAT—have revealed fundamental epigenetic alterations directly associated with impaired lipid metabolism and insulin resistance. Notably, despite the limited correlation between methylomes of different tissues, the identification of conserved epigenetic signatures—such as changes in *NRF1*, *HLA-DQ2*, and *SREBF1*—offers promising opportunities for the development of integrated diagnostic strategies that combine both tissue-specific and systemic epigenetic markers of obesity.

**Table 1 ijms-26-08067-t001:** Methylation biomarkers of early obesity.

Sample, *n*	BMI, Percentile	Detection Method	Material	Biomarkers	Source
28	≥95	HumanMethylation 450K BeadChip	Saliva	*NRF1* (cg01307483)	[53]
94	≥99	MethylationEPIC BeadChip	Peripheral blood	*SREBF1* (cg11024682) *CPA3* (cg13424229)	[37]
16	≥99	MSRE-qPCR	Peripheral blood mononuclear cells	*FTO*	[39]
31	≥95	Methylation-specific PCR	Peripheral blood	*CXCL8*	[38]
190	≥99	HumanMethylation 450K BeadChip, MethylationEPIC BeadChip	Adipocytes of visceral and subcutaneous adipose tissue	*PRRC2A*	[14]
41	≥97	MethylationEPIC BeadChip, Pyrosequencing	Peripheral blood	*TFAM* (cg05831083), *PIEZO1* (cg14926485)	[36]

## 3. The Role of Non-Coding RNAs in the Mechanisms of Obesity Development

Long non-coding RNAs and microRNAs (miRNAs) exhibit a wide range of regulatory functions, influencing processes such as cell differentiation, proliferation, apoptosis, and adaptive stress responses [59]. The execution of these functions directly depends on the biogenesis of non-coding RNAs—a complex and tightly regulated process involving transcription and post-transcriptional processing. A detailed understanding of the biogenetic mechanisms is essential for interpreting the functions of non-coding RNAs and for developing effective strategies for their therapeutic targeting.

MicroRNAs represent a class of small non-coding RNAs, typically about 22 nucleotides in length [60]. Most microRNAs are transcribed from genomic DNA as primary transcripts (pri-miRNAs), which then undergo sequential processing to generate precursor microRNAs (pre-miRNAs) and, ultimately, mature microRNAs [61]. In the canonical mechanism of action, microRNAs interact with the 3′-untranslated region (3′-UTR) of target mRNAs, leading to translational repression. However, evidence also indicates that microRNAs can bind to other regions of mRNA, including the 5′-untranslated region (5′-UTR), coding sequences, and gene promoter regions. Moreover, in certain contexts, microRNAs may exert activating effects on gene expression. Accumulating data highlight the dynamic intercellular transfer of microRNAs, enabling spatiotemporal regulation of both translational and transcriptional activity [62]. Abnormal microRNA expression is associated with the pathogenesis of a wide range of human diseases, including obesity and its related comorbidities [63]. It is important to note that microRNAs are secreted into the extracellular fluid and, thus, can serve as potential biomarkers for various diseases, as well as mediate intercellular communication by functioning as signaling molecules.

Long non-coding RNAs (lncRNAs) are a heterogeneous group of transcripts longer than 200 nucleotides that lack obvious protein-coding potential [64]. The functions of most lncRNAs remain unclear; however, they are known to be involved in the regulation of genomic imprinting, chromatin organization, and allosteric regulation of enzymes [65,66]. Patterns of lncRNA expression coordinate cellular states, differentiation, and development processes. Altered expression of lncRNA genes is associated with various human diseases, including ischemic heart disease and obesity [67,68].

In recent years, accumulating evidence has highlighted the direct involvement of non-coding RNAs, particularly microRNAs, in the pathogenesis of obesity. MicroRNAs can act as paracrine and endocrine messengers and are also involved in the process of adipocyte differentiation [69,70]. Studying microRNA expression profiles in whole blood and plasma is essential not only for understanding the general mechanisms underlying obesity pathogenesis and the impact of adipose tissue alterations on other tissues and organs but also for realizing the clinical potential of microRNAs as biomarkers of future metabolic disorders and therapeutic targets [71].

The study by Tekcan, E. et al. [72] revealed a significant increase in circulating levels of microRNAs miRNA-130b and miRNA-146b in the blood of patients at an early stage of obesity. The biological relevance of these findings lies in the role of miRNA-130b in maintaining energy homeostasis through modulation of the TGF-β signaling pathway, while miRNA-146b directly regulates the expression of *KLF7* (Krüppel-like factor 7), whose overexpression inhibits the activity of critical adipogenic transcription factors. These data suggest that the observed upregulation of miRNA-130b and miRNA-146b may represent a compensatory molecular mechanism emerging in response to impaired adipogenesis during the development of obesity.

Analysis of microRNAs is conducted both in whole peripheral blood and plasma, with each biological material possessing distinct advantages and methodological features. Studies demonstrate that RNA levels in whole blood vary up to 3.4-fold among individuals, reflecting interindividual differences in both cellular composition and RNA content per cell [73]. Peripheral blood, comprising heterogeneous cellular populations (leukocytes, erythrocytes, platelets) and acellular fractions (plasma, serum), provides a comprehensive microRNA profile that reflects the involvement of these molecules in various pathophysiological processes. In contrast to whole blood, plasma has a more homogeneous composition, which enhances the reproducibility of the results but may limit detection of cell-specific microRNAs associated with certain physiological or pathological states. It is noteworthy that individual differences in microRNA expression are influenced by sex and, to a significantly greater extent, by the age of the individual [74].

### 3.1. Circulating MicroRNAs Expression in Plasma

Circulating microRNAs in plasma and serum are molecules with high potential as biomarkers. Their expression levels have been shown to correlate with the development and progression of obesity, which makes microRNAs a promising tool for noninvasive diagnosis and monitoring of obesity-associated pathologies [75].

One of the earliest studies profiling circulating microRNA expression in patients with morbid obesity [76] identified statistically significant alterations in the expression levels of nine microRNAs compared to healthy donors. Five of these—miRNA-142-3p, miRNA-140-5p, miRNA-15a, miRNA-520c-3p, and miRNA-423-5p—were proposed as biomarkers for risk assessment and classification of patients with pathological obesity.

In a 2020 study [77], the role of circulating miRNA-216a in plasma as a promising biomarker of obesity and associated metabolic disorders in the female population was confirmed. The analysis revealed a significant decrease in the expression level of this microRNA in individuals with obesity. Molecular genetic studies demonstrated that miRNA-216a regulates the expression of genes involved in the pathogenesis of obesity, including *PTEN*, *ADIPOR1*, *CAV1*, *CAV2*, and *PPARG*. Epigenetic analysis showed differential methylation of CpG islands in the miRNA-216a locus in patients with early-onset obesity compared to the control group, suggesting a potential mechanism of epigenetic regulation of this gene in the development of metabolic disturbances [78]. The clinical relevance of miRNA-216a is supported by its strong inverse correlation with cardiometabolic risk indicators, including blood pressure (both systolic and diastolic), triglyceride levels, atherogenic index, and markers of systemic inflammation such as high-sensitivity C-reactive protein. Notably, reduced expression of miRNA-216a is significantly associated with increased anthropometric (BMI, waist circumference) and metabolic (blood pressure, triglycerides) parameters [77].

A 2023 study [79] identified that elevated expression of miRNA-15b and miRNA-223 possesses significant diagnostic potential as a biomarker for the early development of childhood obesity and its associated metabolic syndrome. Furthermore, the authors demonstrate the involvement of these microRNAs in key metabolic pathways, including FoxO, insulin, Ras, and AMPK-mediated signaling cascades, highlighting their regulatory role in the pathogenesis of obesity.

### 3.2. Tissue-Specific Changes in MicroRNA Expression

As evidenced by the presented data, miRNAs play a central regulatory role in fundamental adipose tissue processes, including adipogenesis, inflammatory responses, and the development of insulin resistance [80,81,82]. Particular scientific interest lies in the investigation of microRNA expression disturbances at various stages of body mass dynamics, as well as the analysis of changes in their expression profiles in response to therapeutic interventions, which is crucial for the development of personalized approaches to treating metabolic disorders.

A 2023 study [83] conducted a comprehensive analysis of the expression dynamics of microRNAs (miRNA-378a-3p and miRNA-142-3p) in plasma and SAT in adult patients with varying degrees and duration of obesity undergoing sibutramine therapy. The selection of these microRNAs was based on their established roles in the pathogenesis of obesity: miRNA-378a-3p is considered a regulator of adipogenesis during the early stages of the disease, while miRNA-142-3p is viewed as a potential inhibitor of fibrosis in SAT. The study found that baseline expression levels of miRNA-378a-3p did not differ significantly between groups, whereas miRNA-142-3p showed decreased expression in SAT and increased expression in plasma in obese patients. It was established that the expression of both microRNAs correlated with the duration of obesity: peak levels of miRNA-378a-3p were observed during disease onset, possibly reflecting compensatory mechanisms aimed at excess energy accumulation. In contrast, the progression of obesity was associated with a decline in miRNA-142-3p expression, likely driven by pro-inflammatory cytokines and hypoxic conditions. These findings led the authors to conclude that sibutramine may exert an anti-fibrotic effect through activation of miRNA-142-3p.

In the study by Youssef, E.M. et al. [84], the dynamics of microRNA expression in white adipose tissue of Wistar rats were analyzed during diet-induced obesity (high-fat diet, HFD) and subsequent reversion to standard feeding. The results revealed specific time-dependent changes in expression: miRNA-133a showed decreased expression starting from week 6 of HFD, followed by normalization, correlating with glucose levels and supporting its role in adipogenesis regulation; let-7-5p expression progressively declined from week 4, associated with hyperglycemia, hyperinsulinemia, and increased fat mass; miRNA-107-5p exhibited biphasic expression dynamics, with an increase at week 8 (a potential marker of inflammation) and a decrease by week 10 (an adaptive response); miRNA-130a-5p peaked at week 10, possibly mediating inflammatory processes via targeting of *PPARG*; and miRNA-30a-5p showed sustained downregulation throughout the HFD period, likely reflecting ongoing inflammatory alterations. These findings underscore the significant regulatory role of the analyzed microRNAs in the pathogenesis of obesity and highlight their potential therapeutic relevance.

### 3.3. Long Non-Coding RNAs in the Pathogenesis of Obesity

Long non-coding RNAs, defined as transcripts longer than 200 nucleotides, play a regulatory role in modulating gene expression through chromatin remodeling, competitive binding to RNA-binding proteins, and scaffolding of ribonucleoprotein complexes. Modern analytical approaches, including next-generation RNA sequencing and microarray technologies, have enabled the identification of over 3000 lncRNAs in adipose tissue, with approximately 900 transcript variants being specific to brown adipose tissue and associated with improved metabolic status. In patients with early-onset obesity, 1268 differentially expressed lncRNAs have been identified in subcutaneous white adipose tissue, with expression levels correlating with anthropometric and metabolic parameters. Alterations in lncRNA expression profiles in obesity may exert systemic effects on various tissues, potentially contributing to the development of age-related disorders associated with obesity [85]. In metabolic disorders, dysregulation of lncRNA expression is observed, including elevated levels of specific transcripts (*Mist*, lincIRS2, lncRNA-p5549, *H19*, *GAS5*, and *SNHG9*) and reduced levels of others (*Meg3*, *Plnc1*, *Blnc1*, AC092834.1, *TINCR*, and *PVT1*) [86].

In the study by Tang, S. et al. [87], it was demonstrated that overexpression of the long non-coding RNA *Blnc1* (Brown fat lncRNA1) in the epididymal white adipose tissue of mice fed a high-fat diet contributed to a reduction in obesity-associated insulin resistance, partially normalized systemic lipid metabolism parameters, and alleviated hepatic steatosis. Chromatin immunoprecipitation and RNA immunoprecipitation assays revealed that *Blnc1* acts as a competitive target for the RNA-binding protein hnRNPA1, thereby interfering with its interaction with the promoter region of the *Pgc1β* gene and enhancing its expression.

A 2023 study [88] demonstrated that the long non-coding RNA *Hem2atm*, normally highly expressed in M2-polarized macrophages of adipose tissue, shows a significant decrease in expression levels in the subcutaneous adipose tissue of mice with diet-induced obesity. Experimental restoration of *Hem2atm* expression resulted in a substantial reduction in obesity-associated inflammatory processes and improved insulin sensitivity. The molecular mechanism underlying this effect involves the ability of *Hem2atm* to bind heterogeneous nuclear ribonucleoprotein U, thereby inhibiting its stabilizing impact on the mRNAs of pro-inflammatory cytokines (TNF-α and IL-6), consequently suppressing their production.

In the study by Thunen, A. et al. [89], the role of the long non-coding RNA *Lipe-as1* and its isoform *Mlas-V3* in the regulation of adipogenesis in mice was investigated. It was shown that *Mlas-V3*, expressed in the liver and adipose tissue, acts as a negative regulator of the gene encoding hormone-sensitive lipase. Experimental suppression of *Mlas-V3* expression inhibited the differentiation of the OP9 preadipocyte cell line into mature adipocytes, which was accompanied by reduced expression of core adipogenic transcription factors (*Pparg* and *Cebpa*) and increased apoptotic cell death during the differentiation process.

According to the findings reported by Huang, X. et al., the long non-coding RNA *Snhg12*, whose expression is downregulated in obesity, plays a role in suppressing inflammatory responses in adipocytes, enhancing insulin sensitivity, and inducing macrophage polarization toward the anti-inflammatory M2 phenotype [90]. The molecular mechanism underlying this effect involves the ability of *Snhg12* to bind to heterogeneous nuclear ribonucleoprotein A1, which leads to the repression of histone deacetylase 9 expression and subsequent activation of the Nrf2 signaling pathway. Experimental overexpression of *Snhg12* in mice with diet-induced obesity resulted in a significant reduction in inflammation in adipose tissue.

Critical miRNAs and long non-coding RNAs involved in the pathogenesis of obesity are summarized in Table 2 and Table 3.

### 3.4. Methylation of MiRNA Genes in Obesity

MiRNAs, as important regulators of gene expression, themselves are subject to complex multilevel control, necessitating detailed study of the mechanisms regulating their biogenesis and functional activity. Of particular scientific interest is the investigation of epigenetic regulatory mechanisms, specifically the analysis of the influence of DNA methylation patterns in genomic loci encoding microRNAs on their expression activity.

In the study by Mansego, M.L. et al. [78], a significant association was identified between DNA methylation patterns in microRNA-encoding loci (miRNA-1203, miRNA-412, and miRNA-216A) and the development of obesity in childhood. This was manifested by hypermethylation of CpG dinucleotides in miRNA-1203 and hypomethylation of CpGs in miRNA-412 and miRNA-216A in patients with early obesity. It was established that miRNA-1203 regulates the expression of genes involved in the pathogenesis of obesity, including *CIDEA* (a regulator of lipolysis in adipocytes) and *CD44* (involved in inflammation and insulin resistance), and potentially affects the expression of *PLIN4* and *TP53*. miRNA-412 targets *PLIN2*, which plays an important role in the development of hepatic steatosis and inflammatory processes, while miRNA-216A modulates the expression of *PTEN*, a core regulator of insulin signaling, insulin resistance development, and apoptosis.

Thus, the accumulated evidence convincingly demonstrates the critical role of non-coding RNAs in the pathogenesis of obesity, revealing complex mechanisms through which they regulate adipogenesis, inflammatory processes, and metabolic homeostasis. Alterations in the expression of specific microRNAs and lncRNAs across various biological samples, including peripheral blood, plasma, and adipose tissue, not only reflect pathological processes in obesity but also open new prospects for the development of non-invasive biomarkers and targeted therapeutic strategies. Particular importance is given to the study of epigenetic regulation of non-coding RNA genes, which allows for a deeper understanding of the molecular basis of their dysfunction in metabolic disorders. Future research should focus on elucidating tissue-specific effects of non-coding RNAs and developing approaches to modulate their expression for personalized treatment of obesity and its associated complications.

## 4. Chromatin Conformation and Obesity

One of the hallmark features of obesity is the expansion and dysfunction of white adipose tissue, characterized by pronounced adipocyte hypertrophy, impaired adipogenesis, and the accumulation of pro-inflammatory immune cells. To gain deeper insight into the mechanisms driving obesity, numerous studies have investigated genetic factors and identified key genes whose dysregulation may contribute to the development of this condition. However, an increasing body of evidence suggests that obesity is strongly associated with tissue-specific—and even cell type-specific—epigenetic alterations. Epigenomic remodeling in WAT is now considered one of the critical mechanisms linking obesity to its severe clinical complications [91].

The condensation of large genomes within the three-dimensional space of the nucleus is mediated by the formation of dynamic loop architectures. These loops vary greatly in length, ranging from several kilobases (kb) to over a megabase (Mb). Loop connections differ in temporal stability; some loops are nearly constant and persist through most of the cell cycle, while others are transient. It is hypothesized that spatiotemporal variations in loop structures are essential for organizing complex regulatory networks and transcriptional mechanisms [24].

Enhancers and promoters are regulatory DNA elements that facilitate a complex network of intergenic interactions. They are arranged linearly and can regulate the expression of both nearby genes and genes located hundreds of thousands or even millions of base pairs away from the target gene promoters. The interaction between enhancers and promoters positioned at considerable distances is facilitated by the structural loop organization of the genome within the three-dimensional space of the nucleus [92]. Disruption of chromatin looping organization and enhancer–promoter interactions can lead to the development of various pathological conditions, including oncological and genetic diseases, associated with abnormal gene expression caused by the formation of aberrant contacts and activation of alternative promoters [93]. Topological chromatin organization regulates the physical proximity between enhancers and promoters, limiting the range of enhancer action and providing a basis for specific interactions. Enhancer activity is associated with the epigenetic status of the cell and can exist in active, repressed, or poised states depending on the combination of histone marks and other chromatin features [94].

Spatial–temporal control of gene expression depends on the activity of cis-acting enhancer regulatory sequences. These enhancers regulate target genes over various genomic distances and form a complex network (interactome) of interactions between regulatory elements, target genes, and their promoters. It is believed that interactomes are highly specific to particular cell types, enriched with connections between active promoters and active enhancers, and reflect interactions established during cellular differentiation. Thus, during the differentiation of a cell lineage, the three-dimensional genome architecture undergoes stepwise remodeling, involving profound reorganization of promoter–enhancer interactions, changes in chromatin accessibility, and redistribution of histone marks (Figure 2) [95].

### Tissue-Specific Changes of Chromatin Organization in Obesity

A distinctive feature of obesity is the significant enlargement of white adipose tissue, including increased adipocyte hypertrophy, impaired formation of new adipocytes, and accumulation of pro-inflammatory immune cells. Over the past several decades, substantial progress has been made in understanding obesity-induced abnormal remodeling of WAT and its pathophysiology related to obesity-associated metabolic disorders.

Obesity disrupts key functions of human adipose tissue by affecting adipogenesis and the differentiation of preadipocytes into adipocytes. Investigating epigenetic factors that impair adipogenesis is a critical aspect of understanding the mechanisms through which obesity influences normal adipose tissue function. Promoter Capture Hi-C (pCHi-C) methods are employed to identify chromosomal interactions between promoters and their associated regulatory elements. Garske et al. studied interactions in adipocytes between promoters and distant regulatory elements located more than 1 Mb apart (long-range chromosomal interactions, LRI). Such interactions are often overlooked in pCHi-C data analyses due to technical challenges and their low prevalence. The study demonstrated that contact between sequences separated by large genomic distances is reproducible and its frequency increases more than twofold during adipogenesis. Additionally, genomic loci involved in these contacts are epigenetically repressed, correlating with reduced gene expression in LRI regions. Thus, distant interactions marking repressed genomic domains in adipocytes were identified. The authors proposed that these long-range regulatory interactions are enhanced during adipogenesis, highlighting genomic regions that require cell type-specific repression to progress differentiation [96].

Recent evidence indicates that the pathogenesis of obesity is accompanied by progressive adipose tissue dysfunction and the development of chronic inflammation, in which an essential role is played by the infiltration of the tissue by activated macrophages that contribute to the formation of a pro-inflammatory microenvironment [97]. Hata, M. et al. found that obesity in mice, even after its resolution, induces persistent epigenetic changes in macrophage chromatin that promote increased expression of genes involved in inflammatory responses [29]. There is increasing recognition that preadipocytes and adipocytes are also involved in this process, as they are capable of secreting pro-inflammatory cytokines and exhibit gene expression profiles similar to those of macrophages [98]. Garske, K. M. and colleagues [99] investigated the mechanisms by which adipocytes may contribute to the formation of a pro-inflammatory environment. To demonstrate the involvement of epigenetic mechanisms in obesity-associated inflammation, the authors examined primary preadipocytes from pairs of monozygotic twins discordant for BMI, comparing their epigenetic (assay for transposase-accessible chromatin sequencing, ATAC-seq) and transcriptomic (RNA-seq) profiles. They showed that increased BMI affects subnuclear compartmentalization of open chromatin (A compartments) in the preadipocytes of twins, thereby promoting inflammation.

One of the key challenges in studying the mechanisms underlying obesity is the involvement of multiple cell types, tissues, and organs. Chromatin remodeling in response to dietary changes is not limited to adipocytes. With increased intake of high-fat foods, individuals often develop non-alcoholic fatty liver disease (NAFLD). Under conditions of caloric excess, the liver adapts to overnutrition through dynamic transcriptomic reprogramming—an adaptive mechanism to maintain metabolic homeostasis. This reprogramming in hepatocytes is driven by changes in chromatin architecture, which disrupts existing promoter–enhancer interactions and establishes new interactomes, activating genes required for cellular adaptation to the altered environment. Qin, Y. et al. [100] investigated promoter–enhancer dynamics in hepatocyte genomes and showed that chronic high-fat diet intake leads to the activation of genes regulated by peroxisome proliferator-activated receptor alpha (PPARα) and hepatocyte nuclear factor 4 alpha (HNF4α). PPARs are transcription factors activated by fatty acids that regulate energy metabolism and are also expressed in immune cells, where they contribute to cellular differentiation. Both PPARα and HNF4α are constitutively localized in the hepatocyte nucleus and become activated upon ligand binding, with ligand availability modulated by diet and expression of certain genes, such as acyl-CoA thioesterase, which plays a role in lipid metabolism regulation. The authors demonstrated that adaptive processes in response to chronic consumption of a high-fat diet likely influence the activation of main transcriptional regulators. Gene expression is controlled via interactions with promoters through two distinct mechanisms: activation of preformed chromatin loops and the generation of de novo loops. Furthermore, unlike promoter–enhancer loop dynamics, the higher-order chromatin organization—including A/B compartments and TAD boundaries—remained largely unchanged in the liver of the model animals, even in those with markedly different physiological and metabolic profiles [101].

Some studies suggest the existence of depot-specific chromatin architecture, accompanied by variations in chromatin accessibility, reflecting epigenetic plasticity that correlates with transcriptional differences and obesity-related clinical phenotypes [27]. Chromatin accessibility is a defining feature of nuclear hotspots involved in transcriptional regulation and other nuclear processes, characterized by strong enrichment for transcription factor binding sites. Obesity-associated differential gene expression driven by chromatin architecture may not only vary across tissues and cell types but also within anatomically distinct adipose depots.

RNA sequencing experiments revealed distinct gene expression signatures in abdominal fat (AF) and gluteofemoral fat (GF), with 126 genes activated in AF adipocytes and only 90 in GF adipocytes; ATAC-seq and transcriptome profiling showed that 74 of the 126 AF genes (59%) had AF-specific accessible chromatin binding sites, including *HOXA3*, *HOXA5*, *IL8*, *IL1B*, and *IL6*, while only 14 of the 90 GF genes (15%) had GF-specific chromatin accessibility sites (e.g., *HOXC13* and *HOTAIR*), and 25 GF genes (28%) also showed AF-specific open chromatin marks; and ChIP-qPCR confirmed enrichment of the active histone mark H3K4me3 and depletion of the repressive H3K27me3 at *HOXA5* and *HOXA3* promoters in AF preadipocytes, consistent with their AF-specific expression patterns, whereas GF-specific genes *HOTAIR* and *HOXC13* displayed higher H3K4me3 and lower H3K27me3 levels in GF chromatin. Collectively, the authors demonstrated for the first time that adipocytes from anatomically distinct depots exhibit unique gene expression signatures and differential open chromatin profiles [28]. Another study demonstrated that chromatin structure influences the transcriptome of gluteofemoral adipocytes and may be linked to early susceptibility to metabolic syndrome depending on the pattern of obesity [102].

Using a female miniature pig model with diet-induced weight gain and loss, Jin, L. et al. investigated the regulatory mechanisms of three-dimensional genome architecture in adipose tissue. They examined chromatin organization in subcutaneous adipose tissue and three VAT depots. By analyzing transcriptomic changes alongside chromatin architecture remodeling under different dietary conditions, the authors demonstrated that alterations in chromatin structure underlie transcriptomic divergence in adipose tissue, which may be linked to metabolic risk in obesity development. Chromatin structure analysis in SAT cells from different mammalian species suggests the presence of transcriptional regulatory divergence, potentially explaining phenotypic, physiological, and functional differences in adipose tissue. The comparative analysis of conserved regulatory elements between pigs and humans revealed similarities in gene regulation patterns related to the obesity phenotype and identified non-conserved elements within species-specific gene sets that underlie adipose tissue specialization [103].

Weight loss aimed at improving metabolic health and managing comorbidities remains the primary goal of obesity treatment. However, maintaining weight reduction presents a significant challenge, as the body appears to retain an “obesity memory” that resists changes in body mass. Hinte, L.C. et al. [26] demonstrated that both human and murine adipocytes maintain transcriptional alterations for a prolonged period even after substantial weight loss. These findings indicate the existence of epigenetic memory in mouse adipocytes—and likely other cell types—based on stable epigenetic modifications that contribute to obesity persistence. The authors identified significant transcriptional regulatory changes associated with both obesity and post-weight-loss states in adipocytes, adipocyte precursor cells, endothelial cells, epithelial cells, and macrophages compared to other cell types. After weight loss, murine adipocytes sustained activation of regulatory pathways linked to inflammation and the extracellular matrix, whereas adipocyte-specific metabolic pathways remained suppressed, a pattern also observed in human adipocytes. These molecular adaptations seemingly predispose cells to pathological responses within an obesogenic environment, counteracting the effects of weight loss often observed during dietary interventions. Furthermore, the authors revealed marked alterations in chromatin organization across diverse cell populations in obese animal models, alongside increased expression of inflammation-associated genes in adipose tissues following weight gain. Thus, obesity may lead to increased inflammatory responses and/or chromatin remodeling in adipocytes. Targeting these alterations in the future could improve long-term weight management and health outcomes.

## 5. Conclusions

Modern evidence convincingly supports the central role of epigenetic dysregulation in the pathogenesis of obesity, manifested through the complex interplay of three key mechanisms: abnormal DNA methylation patterns in metabolically relevant genes, dysfunction of microRNA regulatory networks controlling adipogenesis and energy homeostasis, and structural chromatin remodeling disrupting the spatial organization of the genome.

Current research in this field reveals a significant correlation between obesity and dysregulation of epigenetic processes, exhibiting pronounced tissue- and cell-specific characteristics. Of particular importance are pathological alterations of the epigenome in white adipose tissue, considered one of the fundamental mechanisms linking obesity to the development of associated metabolic disorders. The study of epigenetics and its role in WAT remodeling remains a relatively young research area; however, it currently attracts substantial scientific interest and is rapidly advancing. The development of novel experimental methods for detecting and analyzing epigenetic modifications has further stimulated interest and progress in this domain. Distinct chromatin architecture within cis-regulatory regions is a crucial factor in governing tissue-specific gene expression and maintaining cellular identity and function.

Promising therapeutic strategies include the development of targeted epigenetic interventions such as antagomirs for suppressing microRNA expression (e.g., RG 125/AZD4076 targeting miRNA-103/107) and agomirs for restoring physiological regulatory networks (agomiRNA-203, which has demonstrated efficacy in correcting metabolic disorders), as well as the use of antisense oligonucleotides and small interfering RNAs to modulate hyperactive signaling pathways. Additional therapeutic potential is offered by histone deacetylase inhibitors (HDACi), which can normalize the metabolic phenotype through regulation of core transcription factors (PPARγ, AMPK, UCP1).

Future research should focus on developing integrated panels of epigenetic biomarkers, creating tissue-specific therapeutic approaches, and investigating transgenerational effects of epigenetic modifications, which requires further in-depth analysis of chromatin architecture and transcriptional networks in various cellular populations of adipose tissue.

## Figures and Tables

**Figure 1 ijms-26-08067-f001:**
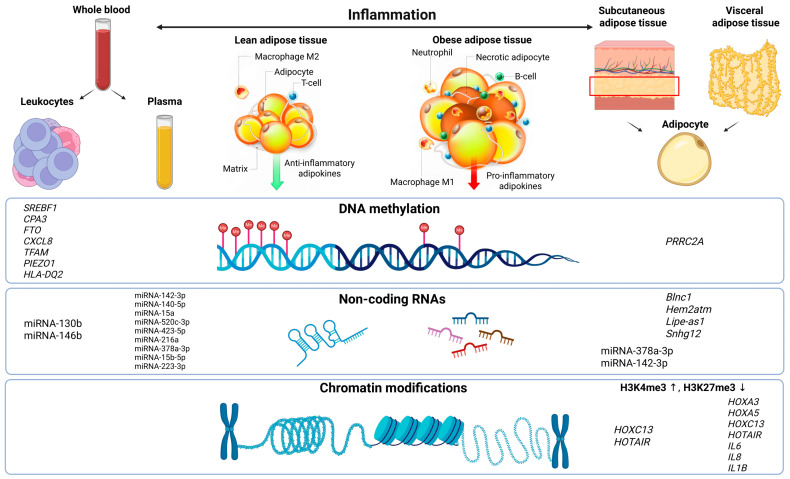
Epigenetic levels of regulation and sources of biomaterials in obesity research. A schematic representation of key sources of biological samples used to study epigenetic mechanisms in obesity and the main biomarkers identified at each regulatory level. On the left—blood and plasma, accessible non-invasive sources for analyzing DNA methylation and non-coding RNAs. On the right—adipose tissue, primarily visceral and subcutaneous, in which epigenetic alterations are detected, including local chromatin modifications and tissue-specific expression profiles of non-coding RNAs. The central panel outlines the major levels of epigenetic regulation: DNA methylation, regulatory non-coding RNAs, and chromatin organization. The top panel illustrates differences between a metabolically “healthy” state and the obese state, including changes in cellular composition and inflammatory status. ↑ denotes increased H3K4me3, while ↓ indicates decreased H3K27me3.

**Figure 2 ijms-26-08067-f002:**
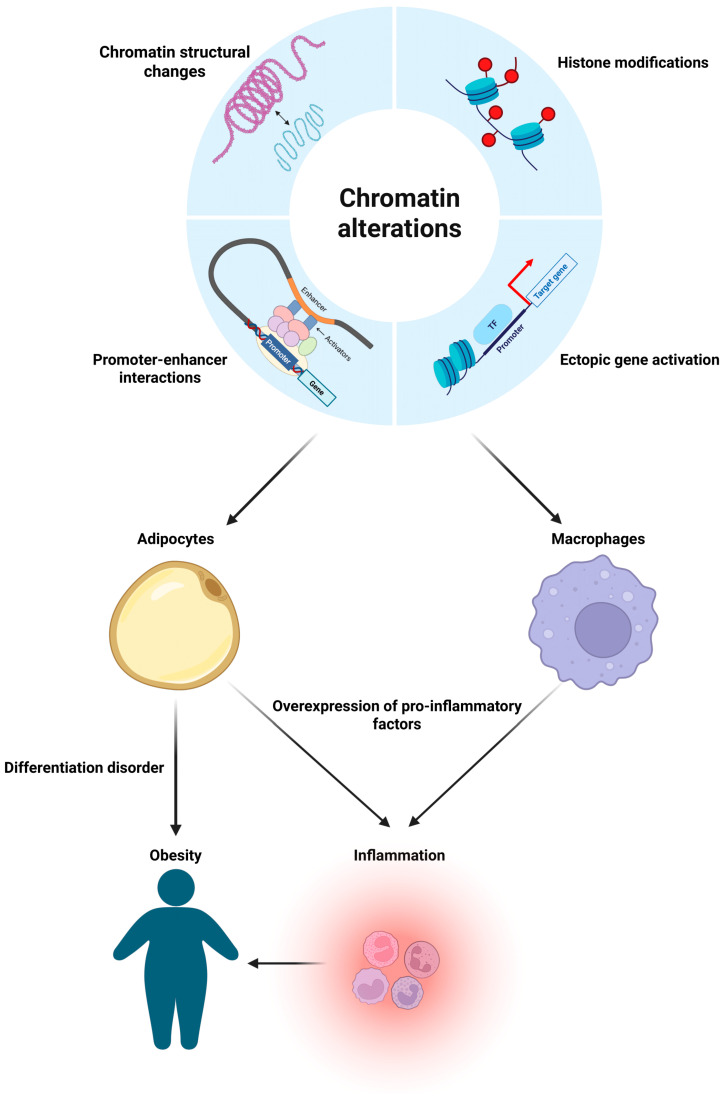
Dysregulation of chromatin represents a key factor underlying the disruption of adipogenesis and chronic inflammation in obesity. Epigenetic modifications, including impaired spatial chromatin organization, dysregulated promoter–enhancer interactions, aberrant activation of target genes, and altered patterns of histone modifications, play a critical role in the pathogenesis of obesity. These modifications lead to the impairment of normal adipocyte differentiation, induction of pro-inflammatory mediators, and adipose tissue macrophage accumulation. The resulting self-perpetuating vicious cycle, characterized by chronic low-grade inflammation and progressive metabolic dysfunction, contributes not only to the maintenance of obesity but also to the development of associated complications, including metabolic syndrome.

**Table 2 ijms-26-08067-t002:** Non-coding RNA expression markers in obesity.

Sample, *n*	BMI, Percentile/ kg/m^2^	Detection Method	Material	Biomarkers	Source
15 (Prepubertal children)	≥95	RT-qPCR	Peripheral blood	miRNA-130b, miRNA-146b	[72]
60 Women	≥30 kg/m^2^	RT-qPCR	Plasm	miRNA-216a	[77]
51 Adults	>30 kg/m^2^	RT-qPCR	Subcutaneous adipose tissue, plasm	miRNA-378a-3p, miRNA-142-3p	[83]
Wister rats	-	RT-qPCR	Subcutaneous adipose tissue	miRNA-133a, let-7-5p, miRNA-107-5p, miRNA-130a-5p, miRNA-30a-5p	[84]
C57BL/6 mice; 3T3-L1 murine preadipocyte culture	-	ChiP, RIP, RT-qPCR	Subcutaneous adipose tissue	*Blnc1*	[87]
HEM2ATM^+/+^ mice	-	RT-qPCR	Subcutaneous adipose tissue	*Hem2atm*	[88]
C57BL/6 mice, OP9 cell culture (CRL-2749)	-	RT-qPCR	Subcutaneous adipose tissue, liver	*Lipe-as1*	[89]
RAW264.7 mice, 3T3-L1 cell culture	-	RT-qPCR	Plasm, liver, adipose tissue, biopsy of the adnexal tissue	*Snhg12*	[90]
12 Children	≥97	HumanMethylation450K BeadChip	Peripheral blood leukocytes	miRNA-1203, miRNA-412, miRNA-216A	[78]
15 Children	>32	Agilent Human miRNA Array V19.0, RT-qPCR	Plasm	miRNA-15b-5p, miRNA-223-3p	[79]

**Table 3 ijms-26-08067-t003:** Key lncRNA regulators of metabolic disorders in obesity (systematized data from recent studies).

LncRNA	Localization	Role in Obesity	Mechanism of Action	Source
*Blnc1*	Brown/white adipose tissue, liver	Regulation of adipogenesis Reduction in insulin resistance Reduction in adipose tissue fibrosis	Activation of PPARγ/C/EBPα Binding to hnRNPA1 → ↑ PGC1β Inhibition of TGF-β	[87,89]
*Hem2atm*	Adipose tissue macrophages (M2)	Suppression of inflammation Improvement of insulin sensitivity	Binding to hnRNP U → ↓ TNF-α/IL-6 Activation of the Nrf2 pathway	[88]
*Lipe-as1 mlas-V*	Adipose tissue, liver	Regulation of lipolysis Control of adipocyte differentiation	Suppression of *LIPE* Regulation of *PLIN4* and *TP53*	[89]
*Snhg12*	Adipose tissue, liver	Anti-inflammatory effect Macrophage polarization toward M2 phenotype	Binding hnRNPA1 → ↓ HDAC9 Activation of Nrf2	[90]

↑ denotes increased PGC1β, while ↓ indicates decreased NF-α/IL-6 and HDAC9.

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
