# Peer review of "When Genes Wear Marks: Epigenomic Modulation in the Development and Progression of Obesity"

_ijms, 2025, doi:10.3390/ijms26168067_

Round 1
Reviewer 1 Report
Comments and Suggestions for Authors
The manuscript details various epigenetic mechanisms that, when modified, may contribute to the progression or development of obesity. This review provides important insights from the authors.
Minor revisions:
- I suggest using the "aberrant" term more discreetly.
- I consider it appropriate to add a section reviewing the expression of miRs in plasma or serum that have been relevant as circulating markers, due to the possibility of using them as easily accessible biomarkers.
- I suggest creating a summary figure for the last section, "chromatin conformation and obesity." Although this is well explained in the text, I believe it is important to highlight the changes and the association between chromatin modifications, their organization, and obesity in a diagram that illustrates this.
Author Response
Thank you very much for taking the time to review this manuscript. Please find the detailed responses below and the corresponding corrections highlighted changes in the re-submitted files.
Comments 1: I suggest using the "aberrant" term more discreetly.
Response 1: Thank you for pointing this out. We agree with this comment. Therefore, we have replaced the term "aberrant" with the more discreet terms "abnormal" and "altered" and highlighted them in red in the text, specifically in lines 13, 141, 225, 298, 329, 339, and 698.
Comments 2: I consider it appropriate to add a section reviewing the expression of miRs in plasma or serum that have been relevant as circulating markers, due to the possibility of using them as easily accessible biomarkers.
Response 2: We agree. Accordingly, we have updated the manuscript by adding a dedicated section (3.1), titled "Circulating MicroRNAs Expression in Plasma" (line 370), to address this point. New microRNAs have also been included in the table 2 and summary figure 1. We have relocated the paragraph discussing "The study by Tekcan, E. et al. [72] revealed..." to the preceding section, now appearing in lines 348-356.
Comments 3: I suggest creating a summary figure for the last section, "chromatin conformation and obesity." Although this is well explained in the text, I believe it is important to highlight the changes and the association between chromatin modifications, their organization, and obesity in a diagram that illustrates this.
Response 3: We have incorporated the suggested content regarding chromatin conformation and obesity into the final section of the manuscript, as presented in Figure 2: "Chromatin dysregulation is a key factor underlying adipogenesis impairment and chronic inflammation in obesity".
Reviewer 2 Report
Comments and Suggestions for Authors
This is an excellent and updated review about the role of several epigenetic mechanisms in the pathogenesis of obesity. Most of mechanisms can be modulated by diet, habits and drugs. It presents a comprehensive analysis of adipocytes and hepatocytes but also in peripheral blood cells. The review has a final section about possible opportunities for the development of novel diagnostic biomarkers and targeted epigenetic therapeutic strategies.
I do not important concerns. The following minor points about the format of the manuscript could be considered.
Fig.1: The list of genes mentioned in "DNA methylation", miRNAs in "Non-coding RNAs" and genes affected by chromatin modifications is not completed. For instance, NRF1 or HLA-DQ2. Why is that? In the same way, Long non-coding RNAs (lncRNAs) are not included at Figure 1, but they should do I suggest that all of them would be listed to give a greater importance to Figure 1 as a summary of the review.
This is a minor point, but reference numbers are frequently far away from the mention, so that sometimes future readers could find difficult to know it. For instance:
Line 162: A Korean study published in 2020..... is that ref. 37?. If so, it is pretty far away (lines beyond 162). I suggest inserting that reference immediately. For instance “A Korean study published in 2020 `[37].....
Line 177: In a study conducted by Rafael S. Lima and colleagues..... is that ref. 38?
Line 267: A recent Latin American study..... is that reference 53?
Line 285: A recent study by Antonio Cantarero-Cuenca….
And so on. Please, look for all examples.
About citations: Family names of the first author could be Ok as used in many paragraphs throughout the manuscript and Table 1. However, at Table 2, the first name is omitted. At Table 3, the number of the reference is just used at the source column. I suggest that the format of the citations would be uniform.
Author Response
Thank you very much for taking the time to review this manuscript. Please find the detailed responses below and the corresponding corrections highlighted changes in the re-submitted files.
Comments 1: Fig.1: The list of genes mentioned in "DNA methylation", miRNAs in "Non-coding RNAs" and genes affected by chromatin modifications is not completed. For instance, NRF1 or HLA-DQ2. Why is that? In the same way, Long non-coding RNAs (lncRNAs) are not included at Figure 1, but they should do I suggest that all of them would be listed to give a greater importance to Figure 1 as a summary of the review.
Response 1: We consider this an excellent suggestion, as we had initially overlooked this opportunity. We have now incorporated the missing markers into Figure 1. However, we would like to clarify that the NRF1 methylation marker was intentionally excluded from Figure 1 because it represents the only saliva-based marker in our review, reported in just a single case, and while its expression in blood has been studied, its methylation status remains unconfirmed.
Comments 2: This is a minor point, but reference numbers are frequently far away from the mention, so that sometimes future readers could find difficult to know it. For instance:
Line 162: A Korean study published in 2020..... is that ref. 37?. If so, it is pretty far away (lines beyond 162). I suggest inserting that reference immediately. For instance “A Korean study published in 2020 `[37].....
Line 177: In a study conducted by Rafael S. Lima and colleagues..... is that ref. 38?
Line 267: A recent Latin American study..... is that reference 53?
Line 285: A recent study by Antonio Cantarero-Cuenca….
And so on. Please, look for all examples.
Response 2: We fully concur with the reviewer's evaluation. In response, we have systematically implemented revisions across the manuscript and have inserted the following references to substantiate connections to original research: line 162 (ref. 37), line 177 (ref. 38), line 210 (ref. 36), line 285 (ref. 58), line 367 (ref. 74), line 376 (ref. 76), line 380 (ref. 77), line 409 (ref. 83), line 424 (ref. 84), line 454 (ref. 87), line 462 (ref. 88), line 471 (ref. 89), line 481 (ref. 90), line 499 (ref. 78), line 604 (ref. 99), line 620 (ref. 100), line 677 (ref. 26).
Comments 3: About citations: Family names of the first author could be Ok as used in many paragraphs throughout the manuscript and Table 1. However, at Table 2, the first name is omitted. At Table 3, the number of the reference is just used at the source column. I suggest that the format of the citations would be uniform.
Response 3: Thank you for pointing this out. We have now standardized all hyperlinks throughout the tables to ensure consistency. We have also standardized the formatting of all first authors' names from the original studies throughout the manuscript text.